Association between phase angle, body mass index and insulin resistance in patients with type 2 diabetes mellitus: a cross-sectional study

Hu Yezi
Jin Hui jinhuison@126.com
Department of Clinical Nutrition, Zhongda Hospital, Southeast University School of Medicine , Nanjing, Jiangsu , China
Hromić-Jahjefendić Altijana
Electronic publication date: 2025 Jan 16
Publication date: 2025
Volume: 13
Electronic Location ID: e18815
Received 2024 May 24; Accepted 2024 Dec 13
Copyright: © 2025 Hu and Jin
Copyright year: 2025
Copyright holder: Hu and Jin
License: This is an open access article distributed under the terms of the Creative Commons Attribution License, which permits unrestricted use, distribution, reproduction and adaptation in any medium and for any purpose provided that it is properly attributed. For attribution, the original author(s), title, publication source (PeerJ) and either DOI or URL of the article must be cited.
License URL: https://creativecommons.org/licenses/by/4.0/

Keywords: Insulin resistance, Phase angle, Body mass index, Type 2 diabetes mellitus, Body composition

Funding: The authors received no funding for this work.

==============================
Background: The purpose of this analysis was to investigate the associations between phase angle (PhA), body mass index (BMI) and insulin resistance (IR) in patients with type 2 diabetes mellitus (T2DM).

Methods: The retrospective cross-sectional study included 200 T2DM patients treated during 2018 to 2019 in Zhongda Hospital Southeast University. PhA and other body composition indicators were measured by bioelectrical impedance analysis (BIA). Subjects were classified into four groups based on body composition: low phase angle and low body mass index (LPLB), low phase angle and high body mass index (LPHB), high phase angle and low body mass index (HPLB) and high phase angle and high body mass index (HPHB).

Results: Overall, in the unadjusted model and minor, all adjusted models (unadjusted model, models 1–4), homeostasis model assessment of insulin resistance (HOMA-IR) was higher in the LPHB group than in the LPLB group (P = 0.034). In the unadjusted model, Model 1 (adjustment for age), Model 2 (adjust for age+duration), Model 3 (adjust for age+duration+sex+UA+TG+TC) and Model 4 (adjust for age+duration+sex+UA+TG+TC+HDL+HbA1c), the adjusted ORs for participants were 4.4 (95% CI [1.72–11.24]), 4.41 (95% CI [1.73–11.27]), 4.75 (95% CI [1.83–12.32]), 2.93 (95% CI [1.04–8.23]) and 3.1 (95% CI [1.09–8.86]) respectively, compared to LPHB group.

Conclusions: T2DM patients with the body composition of low phase angle and high body mass index exhibited the most severe degree and the highest risk of insulin resistance.

Introduction

Type 2 diabetes mellitus (T2DM) has become a major public health concern worldwide (Do et al., 2015). One fundamental problem found in T2DM is insulin resistance (IR), which is strongly related to body composition (Daniele et al., 2014). As T2DM progresses, IR and decreased islet cell function typically occur together (Lin et al., 2020). Therefore, understanding a patient’s body composition becomes crucial in the treatment of T2DM. Although a higher body mass index (BMI) is generally associated with metabolic abnormalities like IR (Fernández et al., 2022), BMI alone fails to differentiate between fat and lean muscle mass, leading to potential misinterpretations. For example, a study by Caan et al. (2018) indicated that a low BMI can mask excess adiposity, while a high BMI can mask low muscularity.

Given the limitations of BMI, alternative measures of body composition are needed. One such measure is the phase angle (PhA) obtained from bioelectrical impedance analysis (BIA), which offers valuable insights into cellular health. To be more specific, PhA can be considered as a marker of membrane integrity and overall cellular function. A higher phase angle indicates healthier cell membranes and greater body cell mass, whereas a lower phase angle may suggest cellular damage, malnutrition, or underlying diseases. Because it is a simple, objective, and non-invasive measure, PhA has increasingly been used to assess nutritional status in various conditions, including diabetes, cancer, and chronic kidney disease (Bučan Nenadić et al., 2022; Shi et al., 2022; Kim et al., 2018). Previous study has found that women with polycystic ovary syndrome (PCOS) tend to have lower PhA values, which may point to a link between low PhA and IR (Barrea et al., 2019). Additionally, research has explored relationship between PhA and factors like uric acid and hand grip strength (Curvello-Silva et al., 2020; De Benedetto, Marinari & De Blasio, 2023). However, there is a lack of information on how PhA, BMI, and insulin resistance are interconnected in patients with diabetes mellitus.

Recognizing the limited information on the interplay between PhA, BMI, and insulin resistance in patients with T2DM, the purpose of this analysis henceforth was to comprehensively investigate these complex relationships. By conducting a cross-sectional analysis, we classified T2DM patients into four groups based on their PhA and BMI: low PhA and low BMI (LPLB), low PhA and high BMI (LPHB), high PhA and low BMI (HPLB), and high PhA and high BMI (HPHB). Our goal was to determine which combinations of body composition are more susceptible to insulin resistance.

Methods

Study design and participants

This cross-sectional, retrospective study analyzed the medical records of 1,098 diagnosed T2DM patients from Zhongda Hospital between January 2018 and December 2019. A flowchart detailing patient selection is presented in Fig. 1. Exclusion criteria were as follows:

1) Age less than 18 years.

2) Severe hepatic or renal abnormalities.

3) Presence of malignancy.

4) Incomplete laboratory tests or bioelectrical impedance analysis (BIA) data.

5) Fasting plasma glucose or fasting serum C-peptide not conforming to the Homeostatic Model Assessment of Insulin Resistance (HOMA-IR) formula—specifically: a. Glucose values less than 3.0 mmol/L or greater than 25.0 mmol/L; b. C-peptide values less than 0.2 nmol/L or greater than 3.5 nmol/L.

Figure 1 Flow chart of participant selection.

Abbreviations: T2DM, type 2 diabetes mellitus; BIA, bioelectrical impedance analysis; FCP, fasting plasma C-peptide.

After applying these criteria, a total of 200 participants were included in this study, consisting of 122 men and 78 women. The “Duration” column in Table 1 indicates the length of time since each patient was diagnosed with T2DM.

Table 1 Clinical characteristics at baseline in the four categories of body compositions classified using PhA and BMI.

Variables	Low PhA
Low BMI
(n = 54)	Low PhA
High BMI
(n = 32)	High PhA
Low BMI
(n = 48)	High PhA
High BMI
(n = 66)	P	
Sex (M/F)	17/37	15/17	35/13	55/11	<0.001	
Age (years)	62.7 ± 13.0	63.5 ± 10.7	52.0 ± 11.1	49.2 ± 14.0	<0.001	
Duration (years)	9.8 ± 7.3	11.8 ± 8.2	8.3 ± 5.7	6.7 ± 6.3	0.003	
TC (mmol/L)	4.7 ± 1.2	4.8 ± 1.3	4.6 ± 1.1	4.8 ± 1.4	0.955	
HDL (mmol/L)	1.2 ± 0.3	1.3 ± 0.4	1.3 ± 0.7	1.2 ± 0.4	0.956	
LDL(mmol/L)	2.9 ± 1.1	2.7 ± 0.9	2.8 ± 0.9	2.7 ± 1.0	0.804	
HbA1c(%)	9.2 ± 1.9	9.5 ± 1.7	9.1 ± 2.2	8.9 ± 1.6	0.561	
FCP (nmol/L)	0.5 ± 0.3	0.7 ± 0.3	0.5 ± 0.4	0.7 ± 0.4	<0.001	
CP120min (nmol/L)	1.8 ± 1.0	2.1 ± 1.0	1.7 ± 1.1	2.6 ± 1.4	<0.001	
FPG (mmol/L)	7.2 ± 1.7	7.7 ± 1.7	7.5 ± 1.9	7.3 ± 1.2	0.544	
HOMA-IR	2.8 ± 0.9	3.5 ± 1.2	3.0 ± 1.3	3.5 ± 1.2	0.004	
BMI (kg/m2)	22.2 ± 2.0	27.8 ± 2.5	23.3 ± 1.3	28.1 ± 2.7	<0.001	
PBF (%)	28.4 ± 7.3	35.4 ± 6.0	23.5 ± 5.3	29.8 ± 5.6	<0.001	
VFA (cm2)	84.2 ± 27.7	136.7 ± 32.7	70.5 ± 17.1	113.3 ± 35.6	<0.001	
PhA (°)	4.4 ± 0.4	4.5 ± 0.4	5.6 ± 0.4	5.7 ± 0.5	<0.001	
UA (mmol/L)	280.5 (212.2, 327.5)	335.5 (257.8, 376.5)	304.5 (254.0, 378.8)	349.0 (297.2, 406.0)	<0.001	
TG (mmol/L)	1.1 (0.8, 2.0)	1.5 (1.2, 2.8)	1.5 (1.2, 2.5)	2.4 (1.6, 4.5)	<0.001	
Notes:

Categorical variables are represented numerically and as percentages. Continuous variables are expressed as the mean and standard deviation (SD) in the case of normal distributions, or the median and interquartile range (IQR) in the case of skewed distributions.

Abbreviations: TC, total cholesterol; HDL, high density lipoprotein; LDL, low density lipoprotein; HbA1c, glycosylated hemoglobin A1c; FCP, fasting C-peptide; CP120min, 120minutes C-peptide; FPG, fasting plasma glucose; HOMA: Homeostasis model assessment; IR, insulin resistance; BMI, body mass index; PBF, percent body fat; VFA, visceral fat area; PhA, phase angle; UA, uric acid; TG, triglyceride.

The study was conducted in accordance with the principles of the Declaration of Helsinki and was approved by the Ethics Review Board of the Zhongda hospital Southeast University (Ethical Application Ref: 2022ZDSYLL302-P01). Due to the retrospective nature of the study and use of anonymized patient data, the requirement for informed consent was waived. This study was registered with the Chinese Clinical Trial Register Network (https://www.chictr.org.cn/, number: ChiCTR2300068513) in 2023.

Measurement of body composition and IR

On the morning of the first day of hospitalization, body composition measurements were conducted using a multifrequency bioelectrical impedance analysis (BIA) device (InBody 770; InBody Co., Ltd., Seoul, South Korea). Tests were performed in the early morning after bowel evacuation to ensure consistency. Subjects stood barefoot on the analyser’s metal footrest with their arms extended downwards, holding the handles in a neutral position. The surface of the hand electrodes contacted each of the subject’s five fingers, while the heels and forefeet rested on the circular foot electrodes.

After an overnight fast of 8–10 h, blood samples were collected to measure fasting plasma glucose (FPG), fasting plasma C-peptide (FCP), plasma lipids, and glycated hemoglobin A1c (HbA1c). High body mass index (BMI) was defined as a BMI greater than or equal to 25 kg/m2, according to the World Health Organization (WHO) criteria. Due to the lack of established cut-off values for PhA, patients were stratified patients based on the median PhA value; PhA values less than or equal to 5.1 were defined as low PhA. All parameters, including PhA and BMI, were directly measured and recorded by the devices. Resistance (R) and reactance (Xc) were used to calculate the PhA by the following equation (Norman et al., 2012):

PhA(degrees)=arctan⁡(XcR)×(180∘π)

IR was determined using the homeostasis model assessment of insulin resistance (HOMA-IR), calculated using the following formula (Shi et al., 2023):

HOMA−IR=1.5+(FPG(mmol/L)×FCP(pmol/L)2,800).

NOTE: In this formula, FCP is expressed in picomoles per liter (pmol/L). Since the FC-P units in our hospital are nanomoles per liter (nmol/L), we converted using the relation 1 nmol/L = 103pmol/L. Patients were classified as insulin resistant if their HOMA-IR value was greater than 2.971. Data were collected as previously described in our published articles (Wang et al., 2023; Hu, Liu & Jin, 2023).

Statistical analysis

Categorical variables are represented numerically and as percentages. Continuous variables are expressed as the mean and standard deviation (SD) in the case of normal distributions, or the median and interquartile range (IQR) in the case of skewed distributions. The chi-square test and Kruskal-Wallis test were employed for the comparison of categorical, normally distributed, and non-normally distributed continuous variables, respectively.

Univariate and multivariable regression analyses were conducted to evaluate the associations between serum IR and measured indicators, in accordance with the recommendations of the Strengthening the Reporting of OBservational studies in Epidemiology (STROBE) statement. Analyses were initially performed without adjustment. Further analyses cumulatively included adjustment for age, duration, sex, uric acid (UA), triglyceride (TG), total cholesterol (TC), high density lipoprotein (HDL) and HbA1c (Model 1–4). Logistic regression models were used for the subgroup analyses and included terms for sex group and age group and the interaction of each subgroup.

All the analyses were performed with the statistical software packages R (R Core Team, 2024) and Free Statistics software versions 1.7.1. A two-sided P value < 0.05 was considered to be statistically significant.

Results

Participants selection

A total of 200 T2DM participants, which consisted of 122 men and 78 women, were classified into four groups according to phase angle and BMI. The baseline characteristics of the participants in the four groups are presented in Table 1. Some differences existed between the four groups with respect to various covariates (age, duration, FCP, CP120min, HOMA-IR, PBF, VFA, UA and TG). However there were no significant differences in TC, HDL, LDL, HbA1c and FPG among the four groups of different body composition. The number of patients in the four groups was 54, 32, 48 and 67 respectively.

In univariate analysis, CP120min, FPG, BMI, PBF, VFA, UA and TG were significantly associated with the IR (P < 0.05) (Table 2).

Table 2 Results of univariate analysis between IR and measured indicators.

Variable	OR 95 CI	P value	
Age	0.99 [0.97–1.01]	0.472	
Duration	0.97 [0.94–1.01]	0.209	
TC	1.09 [0.87–1.36]	0.468	
HDL	1.08 [0.59–1.98]	0.805	
LDL	0.88 [0.66–1.18]	0.391	
HbA1c	0.9 [0.77–1.05]	0.167	
CP120min	2.36 [1.68–3.32]	<0.001	
FPG	1.91 [1.49–2.44]	<0.001	
BMI	1.27 [1.15–1.41]	<0.001	
PBF	1.07 [1.02–1.11]	0.002	
VFA	1.02 [1.01–1.03]	<0.001	
PhA	1.32 [0.9–1.93]	0.151	
UA	1.01 [1–1.01]	<0.001	
TG	1.57 [1.26–1.97]	<0.001	
Note:

Abbreviations: IR, insulin resistance; TC, total cholesterol; HDL, high density lipoprotein; LDL, low density lipoprotein; HbA1c, glycosylated hemoglobin A1c; CP120min, 120minutes C-peptide; FPG, fasting plasma glucose; BMI, body mass index; PBF, percent body fat; VFA, visceral fat area; PhA, phase angle; UA, uric acid; TG, triglyceride; CI, confidence interval; OR, odd ratio.

The ORs and corresponding 95% CIs for the IR according to four groups are summarized in Table 3. Overall, in the unadjusted model and minor, all adjusted models (unadjusted model, models 1–4), homeostasis model assessment of insulin resistance (HOMA-IR) was higher in the LPHB group than in the LPLB group (P = 0.034). In the unadjusted model, Model 1 (adjustment for age), Model 2 (adjustment for age+duration), Model 3 (adjust for age+duration+sex+UA+TG+TC) and Model 4 (adjust for age+duration+sex+UA+TG+TC+HDL+HbA1c), the adjusted ORs for participants were 4.4 (95% CI [1.72–11.24]), 4.41 (95% CI [1.73–11.27]), 4.75 (95% CI [1.83–12.32]), 2.93 (95% CI [1.04–8.23]) and 3.1 (95% CI [1.09–8.86]), respectively, compared to LPHB group.

Table 3 Multivariable-adjusted odds ratios for insulin resistance in the four categories of body composition.

Model	Low PhA+Low BMI	Low PhA+High BMI	P value	High PhA+Low BMI	P value	High PhA+High BMI	P value	
Unadjusted	1 (Ref)	4.4 (1.72–11.24)	0.002	1.31 (0.58–2.94)	0.513	3.28 (1.54–6.97)	0.002	
Model 1	1 (Ref)	4.41 (1.73–11.27)	0.002	1.27 (0.54–2.96)	0.581	3.15 (1.4–7.11)	0.006	
Model 2	1 (Ref)	4.75 (1.83–12.32)	0.001	1.32 (0.56–3.09)	0.525	3.19 (1.41–7.23)	0.005	
Model 3	1 (Ref)	2.93 (1.04–8.23)	0.041	1.1 (0.42–2.88)	0.853	1.59 (0.59–4.26)	0.358	
Model 4	1 (Ref)	3.1 (1.09–8.86)	0.034	1.06 (0.4–2.81)	0.906	1.47 (0.54–3.98)	0.446	
Note:

Model 1: adjust for age.

Model 2: adjust for age+duration.

Model 3: adjust for age+duration+sex+UA+TG+TC.

Model 4: adjust for age+duration+sex+UA+TG+TC+HDL+HbA1C.

Abbreviations: PhA, phase angle; BMI, body mass index; Ref, reference; UA, uric acid; TG, triglyceride; TC, total cholesterol; HDL, high density lipoprotein; HbA1C, glycated hemoglobin A1c.

The stratified analyses were performed to examine whether the association between different categories of body composition and IR was stable among different subgroups. None of the variables, including gender (female and male), age (<65 years and ≥65 years) significantly affected the association between PhA, BMI and IR (all P for interaction >0.05) (Table 4; Fig. 2).

Table 4 Subgroup analysis between PhA, BMI and IR.

Subgroup	Variable	N	OR 95 CI	P value	P.for.interaction	
Sex					0.521	
Male						
	Low PhA+Low BMI	17	1 (Ref)			
	Low PhA+High BMI	15	12.2 [2.24–66.51]	0.004		
	High PhA+Low BMI	35	3.01 [0.76–11.95]	0.118		
	High PhA+High BMI	55	5.6 [1.5–20.94]	0.01		
	Trend.test	122	1.46 [1–2.13]	0.05		
Female						
	Low PhA+Low BMI	37	1 (Ref)			
	Low PhA+High BMI	17	2.97 [0.87–10.15]	0.082		
	High PhA+Low BMI	13	1 [0.23–4.38]	0.997		
	High PhA+High BMI	11	4.2 [0.86–20.61]	0.077		
	Trend.test	78	1.47 [0.91–2.39]	0.119		
Age (year)					0.647	
<65						
	Low PhA+Low BMI	24	1 (Ref)			
	Low PhA+High BMI	18	10.45 [2.44–44.78]	0.002		
	High PhA+Low BMI	44	1.95 [0.64–5.99]	0.242		
	High PhA+High BMI	60	4.54 [1.55–13.34]	0.006		
	Trend.test	146	1.38 [1–1.89]	0.047		
≥65						
	Low PhA+Low BMI	30	1 (Ref)			
	Low PhA+High BMI	14	2.78 [0.71–10.94]	0.143		
	High PhA+Low BMI	4	2.19 [0.25–18.86]	0.475		
	High PhA+High BMI	6	2.01 [0.29–14.15]	0.481		
	Trend.test	54	1.4 [0.77–2.52]	0.269		
Note:

Abbreviations: PhA, phase angle; BMI, body mass index; IR, insulin resistance; Ref, reference.

Figure 2 Stratification analysis on the association between PhA, BMI and IR.

Abbreviations: PhA, phase angle; BMI, body mass index; IR, insulin resistance; Ref, reference.

Discussion

In this study, we examined the relationship between phase angle, body mass index, and insulin resistance in patients with type 2 diabetes mellitus. Our findings revealed a significant association between different body composition profiles and IR, independent of essential covariates and confounders. Notably, it showed Chinese T2DM patients with low PhA and high BMI exhibited the highest degree and risk of insulin resistance compared to other combinations of body composition.

Essentially, PhA can be recognized as an indicator of cell integrity and a nutritional status in various diseases (Grundmann, Yoon & Williams, 2015; Alves et al., 2016; Galluzzo et al., 2018). A lower PhA, reflects poorer nutritional status and worse clinical outcomes. Several studies have reported that individuals with diabetes have decreased PhA compared to healthy controls (Dittmar, Reber & Kahaly, 2015; Jun et al., 2021). PhA has also been closely linked to inflammatory factors (Barrea et al., 2021). For instance, Basile et al. (2014) reported that PhA is positively correlated with muscle strength and mass in elderly patients. Similarly, Kilic et al. (2017) found a negative correlation between PhA and sarcopenia, with an optimal PhA cutoff value of ≤4.55° for detecting sarcopenia: a lack of muscle and a high level of inflammation may lead to insulin resistance (Yoon et al., 2017; Baig et al., 2020).

Certainly, BMI is a good predictor of insulin resistance (Niemann et al., 2020). Overweight and obesity are strongly correlated with IR and T2DM (Tsuda et al., 2018). Indeed, the majority of individuals with T2DM are classified as overweight or obese. Yet IR can vary according to the body composition of individuals with diabetes. Our study demonstrated that T2DM patients who have a combination of low PhA and high BMI experience a more severe degree and a higher risk of insulin resistance compared to those with other body composition combinations.

In T2DM patients, a low PhA often indicates compromised cell membrane integrity and overall, cellular health. PhA may serve as a cell health marker and is closely related to mortality inflammation, nutrition status prognosis and disease outcomes, as it reflects water distribution and body cell mass and representation of cellular integrity. Previous research by our team has found that a lower PhA is a predictor for sarcopenia (Wang et al., 2023). Additionally, a high BMI is often associated with increased adiposity and altered body composition, characterized by higher fat mass and potentially lower muscle mass. When T2DM patients exhibit a low PhA alongside a high BMI, it may signify a combination of factors—such as poor cellular health and excessive adiposity—that contribute to a more severe degree of insulin resistance. While BMI and PhA are both nutritional indicators, higher values are not always better. A high BMI indicates obesity, and excessively high or low PhA values may not be conducive to good health. Therefore, both BMI and PhA should be maintained within a reasonable range.

To our knowledge, this is the first study that explores the association between phase angle, BMI and insulin resistance in T2DM patients. T2DM patients with the combination of low PhA and high BMI experienced the most severe degree and the highest risk of insulin resistance. These results remained consistent across stratified subgroup analyses and demonstrated important clinical implications. To precisely evaluate the susceptibility to insulin resistance and deliver targeted anti-diabetic therapies for individuals with type 2 diabetes, healthcare providers must consider the role of PhA. A simple and non-invasive body composition analysis should be conducted after screening for obesity or overweight using BMI.

Several limitations of the present study should be acknowledged. First, the cross-sectional design does not allow for the establishment of definite causal relationships. Second, the sample size of this study is relatively small, which may limit the statistical power of the findings. Third, we did not account for the presence or absence of diuretic use in patients with cardiovascular disease, yet when reviewing the raw data, none of the patients included had oedema. Lastly, most of the patients were from Nanjing, which could not symbolize the general population in China. However, it is worth mentioning that the interplay between different body composition and IR has not yet been explored clearly.

Conclusions

PhA and BMI are both important body composition markers with T2DM patients. In our study, T2DM patients with the body composition of low phase angle and high body mass index exhibited the most severe degree and the highest risk of insulin resistance. Measurement of PhA may be considered as a routine test for patients with type 2 diabetes in the future.

Supplemental Information

Supplemental Information 1 Raw data.

Supplemental Information 2 Codebook.

We gratefully thank Dr. Jie Liu (Department of Vascular and Endovascular Surgery, Chinese PLA General Hospital) and Dr. Huanxian Liu (Department of Neurology, Chinese PLA General Hospital) for providing assistance in revising this manuscript. We appreciate Mingming Deng for the language polishing.

Additional Information and Declarations

Competing Interests

Author Contributions

Ethics

Data Availability

The authors declare that they have no competing interests.

Yezi Hu conceived and designed the experiments, performed the experiments, analyzed the data, prepared figures and/or tables, authored or reviewed drafts of the article, and approved the final draft.

Hui Jin conceived and designed the experiments, performed the experiments, analyzed the data, prepared figures and/or tables, authored or reviewed drafts of the article, and approved the final draft.

The following information was supplied relating to ethical approvals (i.e., approving body and any reference numbers):

Ethics Review Board of the Zhongda Hospital Southeast University approval to carry out the study within its facilities (Ethical Application Ref: 2022ZDSYLL302-P01).

The following information was supplied regarding data availability:

The raw data are available in the Supplemental File.

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
