# Peer review of "Association between phase angle, body mass index and insulin resistance in patients with type 2 diabetes mellitus: a cross-sectional study"

_PeerJ, doi:10.7717/peerj.18815_

## Round 0.1 · original submission · Major Revisions

Please focus on following points for the improvement (besides reviewer comments):

Language:
The manuscript requires thorough proofreading for grammatical errors and awkward phrasing.
Specific issues include spacing inconsistencies (e.g., lines 44-45 and 57), unnecessary line breaks (e.g., line 99), and unclear sentences (e.g., lines 57-59).
The document should be reviewed by a native English speaker to improve clarity and readability.

Background:
Although the topic is specific, the literature review is lacking. The reviewer suggests including additional relevant references, such as those related to phase angle, BMI, and insulin resistance, to strengthen the review of existing studies.
The authors are encouraged to assess whether suggested references, like the provided semantic scholar link, are relevant.

Article Structure:
Both figures must be resubmitted at a higher resolution to improve visual clarity.
In Table 1, the abbreviations should be moved to the bottom of the table and organized alphabetically or in the order of appearance. Additional explanation is needed for the numbers in brackets in the last two rows (UA and TG).
In Table 2, abbreviations should be reordered for consistency.

Experimental Design:
Clarification is needed on what the variable "duration" represents in the methods section.
The inclusion criteria are a significant concern. Patients with cardiovascular disease should be excluded from the study due to their use of diuretic drugs, unreliable fluid status, and edematous conditions, which may impact the findings.

Conclusions:
The conclusions section is too brief and should be expanded to include a summary of the key findings from the study, rather than presenting only a single conclusion.

·

Basic reporting

The paper excellently shows an example of the connection between phase angle body mass index and insulin resistance in patients with 2 type 2 diabetes mellitus

Experimental design

Well stated and explained.

Validity of the findings

No comment

Additional comments

The article is well-written and clear.

Reviewer 2 ·

Basic reporting

### Language
The document should be proofread and the mistakes should be corrected.
E.g. unnecessary space at the end of the abstract (no line number).
Lines 44 and 45: space is missing after the end of the sentence.
Line 57: “has shown”
Line 85: “Measurement of…”
Line 99 has an unnecessary line break.
There are many other mistakes. Moreover, certain sentences are not as comprehensive as they could/should be – e.g. the sentence on the lines 57-59. Again – it is advisable that the manuscript is proofread by a native speaker.
### Background
The topic of the article is specific and it seems that the specific relationship between phase angle, body mass index, and insulin resistance is not studied extensively, which is why the authors do not have a lot of literature review. However, I believe there are still references that could be included, e.g. https://pdfs.semanticscholar.org/e7a5/3fb3223f051921bbc574adeeaade946ef30f.pdf.


### Article structure
Both figures are of low quality/resolution and should be resubmitted at a higher/satisfactory resolution.
Table 1: abbreviations should be moved to the bottom and ordered either alphabetically or according to the order of their appearance in the table. The last two rows (UA and TG) should be further explained – what do numbers in brackets represent – e.g. 280.5 (212.2,327.5).
Table 2: abbreviations should be ordered.

Experimental design

### Methods described
The authors do not explain in the article what does the variable “duration” represent.

Validity of the findings

### Conclusions
“Conclusions” section should be expanded to include more than one conclusion – briefly report the main results of the study.

Additional comments

Overall, the study is well-designed, with clearly defined goals, methods, and results. The raw data submitted is clean with no apparent issues. Minor technical things should be changed.

Reviewer 3 ·

Basic reporting

Upon reviewing the manuscript titled "Association between phase angle, body mass index and insulin resistance in patients with type 2 diabetes mellitus: a cross-sectional study," I have identified several areas for improvement:

1. The manuscript requires thorough English language editing.

Experimental design

2. The inclusion criteria raise concerns regarding the exclusion of patients with cardiovascular disease. It is essential to exclude these patients due to their use of diuretic drugs and unreliable fluid status, as well as the presence of edematous conditions.
3. The pre-measurement conditions for calculating the Phase angle in patients are not clearly outlined, and this is a critical aspect that needs to be addressed.

Validity of the findings

4. The manuscript lacks a clear final message, and the findings do not appear to be groundbreaking, particularly considering the confounding factors and the small sample size of the diabetic patient population.

---

## Round 0.2 · Major Revisions

Table 1 is still missing the explanation on intervals. I suppose these are standard deviation and confidence intervals, but it is not written (like table two, e.g., clearly stating 95CI).

Moreover, several grammatical errors require attention. Additionally, the text would benefit from English language polishing.

Regarding the measurement with bioimpedance, it is imperative to conduct pre-measurements as per the requirements outlined in the previous arbitration. Unfortunately, the authors did not provide a convincing answer, and the text did not adhere to these stipulations.

Lack of explanation regarding the use of drugs, particularly diuretics and antihypertensive drugs, and their correlation with the findings.

Reviewer 2 ·

Basic reporting

### Language
The document was thoroughly checked and rewritten.

### Background
Additional literature has been added in the background.

### Article structure
Better-quality images have been submitted and tables have been edited. Table 1 is still missing the explanation of intervals. I suppose these are standard deviation and confidence intervals, but it is not written (like table two, e.g., clearly stating 95CI)

Experimental design

### Methods described
"Duration" has been explained

Validity of the findings

### Conclusions
Conclusions have been expanded.

Additional comments

Everything has been implemented except for the small detail: Table 1 is still missing the explanation on intervals. I suppose these are standard deviation and confidence intervals, but it is not written (like table two, e.g., clearly stating 95CI).

Reviewer 3 ·

Basic reporting

After a careful review of the manuscript, it shows that several grammatical errors require attention. Additionally, the text would benefit from English language polishing.

Experimental design

Regarding the measurement with bioimpedance, it is imperative to conduct pre-measurements as per the requirements outlined in the previous arbitration. Unfortunately, the authors did not provide a convincing answer, and the text did not adhere to these stipulations.

Validity of the findings

Furthermore, there is a lack of explanation regarding the use of drugs, particularly diuretics and antihypertensive drugs, and their correlation with the findings.

Additional comments

It is also important to note that the discussion on dissection is inadequate and should not be linked to the phase angle with the risk of developing organ function disorders.

---

## Round 0.3 · Minor Revisions

Please add a more descriptive legend for Table 1, as to what the various items represents. We suppose these are standard deviation and confidence intervals, but it is not written in the legend

Reviewer 2 ·

Basic reporting

Nothing new.

Experimental design

Nothing new.

Validity of the findings

Nothing new.

Additional comments

Nothing new.

---

## Round 0.4 · accepted · Accept

Your article is accepted.